# Does the Incorporation of Biochar into Biodegradable Mulch Films Provide Agricultural Soil Benefits?

**DOI:** 10.3390/polym16233434

**Published:** 2024-12-06

**Authors:** Kesinee Iamsaard, Nuttapon Khongdee, Raweerat Rukkhun, Charoon Sarin, Pantip Klomjek, Chanin Umponstira

**Affiliations:** 1Department of Natural Resource and Environment, Naresuan University, Phitsanulok 65000, Thailand; kesineei@nu.ac.th (K.I.); charoons@nu.ac.th (C.S.); pantipk@nu.ac.th (P.K.); chaninum@nu.ac.th (C.U.); 2Department of Highland Agriculture, Chiang Mai University, Chiang Mai 50200, Thailand; 3Faculty of Natural Resources, Prince of Songkla University, Songkhla 90110, Thailand; jureerat.ru@psu.ac.th

**Keywords:** biodegradable mulch film, biochar, soil carbon, systematic review

## Abstract

The pollution caused by plastic mulch film in agriculture has garnered significant attention. To safeguard the ecosystem from the detrimental effects of plastic pollution, it is imperative to investigate the use of biodegradable materials for manufacturing agricultural plastic film. Biochar has emerged as a feasible substance for the production of biodegradable mulch film (BDM), providing significant agricultural soil benefits. Although biochar has been widely applied in BDM manufacturing, the effect of biochar-filled plastic mulch film on soil carbon stock after its degradation has not been well documented. This study provides an overview of the current stage of biochar incorporated with BDM and summarizes its possible pathway on soil carbon stock contribution. The application of biochar-incorporated BDM can lead to substantial changes in soil microbial diversity, thereby influencing the emissions of greenhouse gases. These alterations may ultimately yield unforeseen repercussions on the carbon cycles. However, in light of the current knowledge vacuum and potential challenges, additional study is necessary to ascertain if biochar-incorporated BDM can effectively mitigate the issues of residual mulch film and microplastic contamination in agricultural land. Significant progress remains necessary before BDM may fully supplant traditional agricultural mulch film in agricultural production.

## 1. Introduction

Agricultural plastic film is an effective cultivation material to maintain soil temperature and soil water, reduce pesticide usage, prevent soil erosion, and suppress weed proliferation [1]. Non-biodegradable petroleum-based films such as polyethylene (PE), followed by polypropylene (PP), polyvinyl chloride (PVC), ethylene vinyl acetate (EVA), polymethyl methacrylate (PMMA), and polycarbonate (PC) have been widely utilized in agriculture leading to a significant environmental problem [2]. The main cause of this issue is the exhaustion of fossil fuels used in the production of polymers, which leads to a higher amount of plastic trash being generated, subsequently contaminating the natural environment with microplastics. Microplastics can lead to a decrease in soil aggregation and soil bulk density while also increasing the rate at which soil water evaporates [3]. The accumulation of agricultural plastic residues has become a concealed threat to the quality and safety of agricultural soil, obstructing gas exchange between soil and water, distributing water-stable aggregates, bulk density, water retention capacity, and pH value, and modifying the soil microbial population [4,5]. In China, the utilization of agricultural film escalated from 600,000 tons in 1981 to 137.9 million tons in 2019, representing a 230-fold growth. The prediction is that plastic film usage will reach 228 million tons by 2025 [2]. Hence, it is crucial to conduct additional investigations into biodegradable materials used for agricultural films in order to reduce plastic mulch pollution.

The residual plastic mulch adversely impacts soil structure, characteristics, and moisture absorption, leading to the unsustainable use of agricultural land. BDMs can be synthesized from biobased polymers obtained from microorganisms, plants, or fossil-derived substances [6]. Prevalent biobased polymers utilized in BDMs comprise polylactic acid (PLA), starch, and cellulose. Polymer derived from fossils utilized in BDMs comprise poly (butylene succinate-co-adipate) (PBSA), poly (butylene adipate-co-terephthalate) (PBAT), poly (butylene succinate) (PBS), poly (caprolactone) (PCL), and poly (vinyl alcohol) (PVOH) [7,8,9]. Polymers utilized in BDMs have ester linkages or polysaccharides, which are susceptible to microbial degradation into carbon dioxide, methane, water, and microbial biomass [6,8]. Certified soil-biodegradable mulch films adhere to international criteria (e.g., EN 17033:2018 [10]) that guarantee their effective decomposition in agricultural environments without residual impact [11]. Employing BDM can improve soil physicochemical and microbiological properties and crop productivity, with certain biodegradable films exhibiting performance comparable to PE plastic films [12,13,14,15]. However, it is premature to advocate for the widespread use of BDM without definitive proof of their potential ecotoxicity to soil ecosystems [16]. Particularly, the effect on soil processes, including carbon sequestration, remains predominantly unexamined [8]. Additionally, the drawback of BDMs remains a topic of discussion concerning the expense of biopolymers, which constitute the primary component in BDM production. Widely utilized biopolymers, such as PLA, necessitate a production cost of over USD 4000 per metric ton, whereas traditional polymers are priced at roughly USD 1000 to 1500 per meter ton. Thus, BDMs are approximately 1.5 to 1.8 times more costly than plastic mulches [17].

One alternative approach for reducing the cost of biopolymer manufacture is to incorporate organic materials as a natural filler in the BDM. Natural fillers in biodegradable composite films not only alleviate the detrimental effects of synthetic materials on the environment, such as incomplete decomposition but also possess advantageous characteristics, including renewability and high specific strength, rendering them suitable for the production of composites for diverse applications [9,18]. The creation of materials yielding films with unique properties, such as biodegradability and non-toxicity, is appealing due to their extensive applicability and significantly reduced environmental impact [19]. Novel materials for product deterioration via the enzymatic activity of bacteria, yeasts, and/or fungi allowing for usage as fertilizers and soil conditioners are required [20]. Several studies have reported the filling of natural material in BDMs for improving their properties, such as empty fruit bunches [21], cotton fibers [22], chitosan [23], alginate [24], starch [25], and cellulose [26]. Previous studies have reported that BDM can improve the function of soil conditioners and productivity. A biodegradable mulching sheet containing the highest concentration of urea significantly enhances seedling growth [18]. A separate study indicated that the decomposition of carbon waste ash-reinforced starch films can release the nutrients contained in the ashes into the soil [27].

As previously explained, biochar is able to serve as an alternative natural material for manufacturing the BDM. It is posited that the breakdown of biochar composites with biocomposite-based mulching films can enrich the soil with nutrients following decomposition. The biochar-derived biocomposite presents a viable option for producing BDM due to its elevated biodegradability rate and substantial nutrient content while addressing the issue of plastic waste contamination. Biochar is a highly aromatic porous carbonaceous material produced through the thermochemical conversion of organic material (such as agricultural waste) in oxygen-limited conditions. Initially, the utilization of biochar as a soil remediation agent has been firmly established in the field of agriculture. These uses encompass enhancing the composition and productivity of depleted soils, retaining soil moisture, and sequestering carbon [28,29]. Biochar possesses distinctive chemical, physical, and biological properties, making it a versatile material with a wide range of applications [30]. Over the past few decades, interest in converting biomass into biochar has grown significantly, driven by its multiple benefits and diverse application potential [31]. As biochar matures, it becomes part of soil aggregates, safeguarding the carbon in biochar and facilitating the stabilization of rhizodeposits and microbial products. Biochar carbon remains in the soil for extended periods ranging from hundreds to thousands of years [32]. The physiochemical properties of biochar, including its three-dimensional reticulated and porous structure, make it an effective long-term carbon storage solution that can also absorb and degrade contaminants. The main benefits of biochar-based materials lie in their highly porous, large surface area, better ion exchange capacity, and plentiful functional groups [33]. The numerous features of biochar can enhance the functionality of BDM. Biochar’s non-graphitic structure is rich in surface functional groups such as C–O, C=O, COOH, OH, etc. These groups have the potential to create additional chemical interactions with a polymer matrix [28]. Considering these properties, the application of biochar as a filler can improve the properties of composites and broaden their usability. For example, it has been reported that biochar improves the composites’ tensile modulus of elasticity, as well as the strength of Hemp-PLA composites [34], and enhances mechanical properties, thermal transitions, and biodegradability poly (butylenesuccinate) (PBSu) [35]. Biocomposites incorporating 1%, 2.5%, and 5% biochar were synthesized utilizing PBSu. The incorporation of biochar enhanced mechanical characteristics and temperature transitions while also improving biodegradability [35]. PLA infused with dairy manure exhibited decreased crystallization during the heating process in the differential scanning calorimetry, contrasting with the behavior of PLA infused with wood chip biochar. Consequently, wood chip biochar emerged as the optimal filler for PLA [36]. A study by Merino et al. [37] reported that lemon peel biochar was introduced at a 5 wt% and exhibited mechanical qualities suitable for mulching. The rapid biodegradability in soil with a 50% weight reduction was seen within the initial week of the experiment. Additionally, visual assessment revealed that the augmentation of wood chip biochar content resulted in an increase in cracks on the surfaces of the PBAT/PLA/biochar composites [38]. Biochar composite with BDMs for various functions has now been increasingly studied, including as a fertilizer carrier [39]. The ongoing controversy concerning BDMs pertains to their partial degradation in agricultural environments. Although biochar aims to improve biodegradability, questions arise regarding its lingering impacts if these films fail to decompose completely, perhaps resulting in microplastic pollution in soils [6,40,41]. Studies reported that increased biochar content can encapsulate degradation products inside the polymer matrix, potentially decelerating the overall degradation process. Composites containing 20–30% biochar demonstrated retarded degradation owing to the hydrophobic characteristics of biochar and its capacity to encapsulate components inside the matrix [42]. The durability of biochar in soil is significant, as it decomposes at an exceedingly slow rate, with mean residence lengths projected to span hundreds of years. The gradual breakdown may enhance the longevity of biochar in biodegradable composites, thus influencing their degradation characteristics [43,44]. Furthermore, environmental factors such as temperature and microbial activity significantly influence the rate of decomposition of these materials [45].

The emission reduction commitments outlined in the Paris Agreement have been enhanced on a worldwide scale since 2020. Reducing greenhouse gas (GHG) emissions and storing carbon are two essential methods to address global warming [29]. Soil, which is the greatest reservoir of organic carbon (OC) on land, contains a greater amount of OC than the total amount found in global plants and the atmosphere [46]. Nevertheless, the majority of soil organic carbon (SOC) pools, including forests, permafrost, and wetlands, are not actively controlled or manipulated. Only agricultural soil has the potential to be actively managed in order to enhance carbon sequestration [47]. Therefore, the sequestering OC in agricultural soil has gained significant societal and scientific interest because of its significant impact on soil health and the mitigation of climate change [48]. Increased soil carbon sequestration in agriculture can be accomplished by many management techniques, such as implementing cover cropping, practicing no-tillage, rotating crops, and incorporating organic matter. These strategies promote plant development and enhance soil microbial activity. These mechanisms result in the breakdown of stable carbon, thus preventing its emission into the atmosphere [49].

The alternate approach for producing BDM based on enhanced soil quality has garnered significant interest. A rationale for employing biochar-composite-based BDMs to enhance soil carbon reserves is presented. Biochar integrated into BDM could enhance soil carbon sequestration during its degradation. Utilizing biochar enhances the carbon content in the soil, since it represents a stable form of carbon that endures for prolonged durations. This stability aids in preserving elevated carbon stores throughout time, which is essential for alleviating climate change by diminishing atmospheric CO_2_ concentrations [50,51]. Although mulching may occasionally elevate greenhouse gas emissions due to increased microbial activity, the use of biochar composite in BDM could alleviate these impacts. Increased biochar application rates correlate with decreased emissions of carbon dioxide and nitrous oxide. This dual strategy enhances crop yields while also reducing the overall carbon footprint of agricultural activities [52,53].

As mentioned above, filling natural material is necessary to enhance the practical application of BDM. While numerous studies have examined the role of biochar in enhancing the mechanical properties of BDMs, limited research has demonstrated the practical application of biochar composite on BDMs for the soil carbon stock contribution after its degradation. The debated issue of soil carbon sequestration undertreated with BDM remains largely unstudied. To date, a few studies have reported the application of biochar integrated with agricultural mulch film to enhance soil carbon [54]. Thus, the objective of this review is to investigate the efficacy of biochar incorporated in BDMs in enhancing soil carbon sequestration in agricultural soils. Two key inquiries were addressed: (1) the application of biochar integrated with various bioplastic mulch films and (2) the potential contribution of biochar incorporated in BDM to soil carbon sequestration. This study provides insights into BDMs incorporated with biochar as an alternative material in agricultural production for reducing greenhouse gas emissions. This review emphasizes the theoretical and empirical findings regarding the physical and chemical impacts of biochar-derived BDM on soil carbon sequestration. This evaluation would facilitate the progression of sustainable materials and the enhancement of environmental consciousness.

## 2. Quantitative Assessment of the Publications

The annual publication count was aggregated to analyze the trend of BDM research, as illustrated in Figure 1. At the beginning of the investigation, only 13 articles related to BDM were published in 2001. The research initially concentrated on the characterizations of as-produced BDM [55]. The increase in publications ten years later accounted for 50% compared with those from the beginning, while the development of BDM by incorporating natural materials into BDM has been happening since 2005. Two decades later, the study viewpoints were varied, encompassing natural material filler [27,56] and biodegradation validation on the soil environment [7]. A significant rise in publications was noted in 2024, reaching up to 56 times that of 2001. Interestingly, the quantity of BDM publications rose from 502 to 738 within a single year (2023 to 2024). Even the environmental impact, such as carbon footprint analysis under BDM, was highly investigated [14,57,58]. Therefore, the figure indicates a substantial rise in the number of publications post-2015, particularly from 2020 onwards, concerning the practical application of BDM on soil functions. This probably directs an increasing interest in advancing technology pertinent to soil management and environmental sustainability. The evaluation of environmental impact, particularly on the soil environment of BDM, which directly affects sustainable agricultural development, has garnered significant attention from researchers. Also, the advancement of research methodologies facilitated the evolution of BDM [16]. However, recent inquiries investigating the effects of biochar-enriched BDM on soil carbon have received diminished attention from 2024 onwards. Possible reasons may be due to the intricate relationships between biochar, BDM, and soil microbial activities. These complexities present challenges for researchers, potentially leading to reduced interest in this specific area of study [45]. Additionally, concerns surrounding microplastic contamination may stem from the insufficient degradation of natural material fillers, including biochar, in biodegradable mulch films used in agriculture. Biodegradable mulch films, although intended to degrade over time, may leave residues that contribute to microplastic pollution if they fail to fully break down. Studies demonstrate that biodegradable polymers may still release microplastics during degradation, especially under suboptimal environmental conditions for total disintegration [59,60].

The literature study indicates that the potential benefits of utilizing biochar as an additive in BDM have been reported since 2015 [61,62]. The documentation outlines the possible benefits of incorporating biochar as an ingredient in BDM products, highlighting its advantageous properties such as elevated surface area and enduring chemical and physical stability. Documented enhancements in the performance of polymer–biochar composites encompass increased water adsorption, heat resistance, and rigidity [63]. Despite the documented use of biochar as an addition in BDM since 2015, current investigations into the impact of biochar-enriched BDM on soil carbon have garnered less attention.

## 3. Comparative Assessment of BDM Degradation in Soil

Figure 2 illustrates the comparison between the source and degradation of two types of polymers: bio-based and fossil-based polymers. Based on the source of the raw materials and their environmental impact, all bioplastics may be classified into three groups [64]. Initially, it is a non-biodegradable polymer created from bio-based sources. The second group comprises biodegradable polymers sourced from either bio-based or fossil-based materials. The final category consists of a non-biodegradable material originating from fossil-based sources. Table 1 presents a comparative analysis of the advantages and limitations of bio-based polymer, BDM, and other fossil-based films (such as PET, PP, PS, PVC, PE, etc.) from prior research [6,7,65,66,67,68,69,70]. BDMs have considerable ecological benefits compared with conventional fossil-derived films such as PE, chiefly due to their capacity for biodegradation and their positive impact on soil health. Nevertheless, they encounter obstacles, including elevated expenses and restricted market acceptance. Conversely, fossil-derived films are more economical and prevalent; however, they exacerbate enduring environmental challenges.

BDM is not pure polymers; rather, they require degradation in an agricultural setting through the activity of indigenous microbes, some of which may or may not degrade under specific environmental conditions [71]. Thus, to examine the practical application of BDM concerning degradation in the soil environment, a synopsis of the bioplastic type is an essential investigation. The literature analysis indicates the summary of polymer-based agricultural mulch production and its corresponding rates of biodegradation in soil, as shown in Table 2. Biodegradability under typical environmental conditions is not contingent upon the polymer’s source; rather, its chemical structure and physical qualities are critical determinants [7]. Plastic mulch films have been replaced with BDM. These mulches should be tilled into the soil after usage so that natural microbes can break down the plastic. BDM can be made from either biobased polymer generated by plants or microbes or fossil-derived materials [71]. The category of biodegradable polymers comprises bio-based materials, including cellulose, starch, PLA, and poly (hydroxyalkanoates) (PHAs). Biodegradable polymers originating from fossil-based sources encompass PBAT, poly (butylene succinate) (PBS), PBSA, PCL, and PVOH. When evaluating the biodegradable polymer derived from renewable resources, its primary advantage is in its ability to replenish the carbon cycle, as the duration required for production and conversion to biomass is comparable. Biodegradable polymers derived from bio-based resources convert to biomass significantly more rapidly than fossil-based polymers, which require millions of years for the same process [7]. However, mulch film properties cannot be predicted based on the properties of pure polymers. A study by Arias et al. [72] and Gattin et al. [73] reported that the degradation behavior of PLA was modified upon blending with PHB and other additives, leading to alterations in the miscibility of the polymer components. To combat plastic pollution, cradle-to-cradle strategies focused on the creation of highly recyclable and biodegradable polymers with minimal environmental impact are gaining traction [15,74]. One approach to mitigate the greenhouse gas emissions generated by plastics is to substitute fossil-based plastics with bio-based alternatives [75].

## 4. Biochar–Bioplastic Composite in Biodegradable Mulch Film

The utilization of biochar as a versatile filler in BDM has garnered significant interest from the scientific community in recent years, owing to its remarkable potential for creating sustainable and high-performance materials. The primary advantages of utilizing biochar as a filler in the fabrication of BDM pertain to the enhancement of mechanical properties, electrical conductivity, and thermal stability, facilitated by the incorporation of a sustainable and renewable material [85]. Most polymer composites consist of a thermoset or thermoplastic matrix combined with organic fillers such as PHA, PLA, and PHB [9].

Table 3 presents a summary of prior articles about the development of biochar bioplastic composites for agricultural mulch films. Biochar has been utilized as an ingredient in BDM items owing to its advantageous properties, such as elevated surface area and enduring chemical and physical durability. Documented enhancements in the efficacy of polymer–biochar composites encompass increased water adsorption, heat resistance, and stiffness. The advantages of preventing organic waste disposal in landfills (which may produce methane emissions) and sequestering carbon inside the biochar material enhance its appropriateness for incorporation into circular manufacturing systems [63].

Biodegradable plastic has emerged as a favored substitute for traditional plastic in response to the plastic pollution challenge. The superior mechanical qualities of bioplastics enable their application in various potential areas, including agriculture. Nonetheless, bioplastics are infrequently selected as the primary material due to their significantly elevated cost. Consequently, a cost-effective natural filler, readily obtainable from agricultural waste, is suggested for incorporation into the polymer to create a biocomposite [18]. The potential of food-waste-derived biochar as a filler material has its associated problems, including inadequate dispersion and heightened thermal degradation in PLA. A significant discovery of this study is that biochar produced from food waste enhanced the degradation rate of PLA under composting conditions, exhibiting nearly double the mass loss after 40 days in samples with high biochar content compared with pure PLA [86]. The use of biochar resulted in an enhancement of the elastic modulus while preserving elevated deformation values. Measurements of the water contact angle indicated an enhancement in the hydrophobic properties of the biocomposite films relative to PBAT. Furthermore, accelerated deterioration testing, monitored through tensile tests and spectroscopic analysis, demonstrated that the filler conferred photo-oxidative resistance to PBAT by postponing degradation processes [87]. The potential of biochar generated from various biomass residues, including cassava rhizome, durian peel, pineapple peel, and corncob, as a filler or reinforcement to enhance the mechanical characteristics of polylactic acid (PLA) has been assessed. Among these biomass residues, carbon-rich biochar obtained from durian peel showed significant improvement in the mechanical properties of PLA-biochar composites. The tensile strength and elongation at the break of the PLA composite diminished upon the incorporation of durian biochar, decreasing from 14.9% to 10.4%. This may be attributed to the inefficiencies in stress transfer and the uneven distribution of biochar particles inside the matrix [88].

However, the ambiguous present circumstances regarding the incorporation of biochar in AFM are currently being addressed. The cost of producing BDMs has been high. Additionally, elevated manufacturing costs have consistently been a significant constraint for biopolymers. Widely utilized biopolymers, such as PLA, necessitate a production cost of over USD 4000 per metric ton, whereas traditional polymers are priced at roughly USD 1000 to 1500 per meter ton. Consequently, BDMs are roughly 1.5 to 1.8 times more expensive than plastic mulches [17]. A comprehensive techno-economic analysis and life cycle assessment indicated that biochar is currently not a viable choice in film production. Biochar formulas necessitated increased thickness, adversely affecting both the cost and environmental impact of the film [54]. The promotion of biodegradable films encounters several challenges, namely high costs, farmers’ reluctance to use them, and difficulties in their promotion. The government ought to cultivate and leverage ample and cost-effective biological resources tailored to local conditions while establishing film production enterprises to minimize transportation expenses, thereby decreasing unit prices and increasing farmers’ propensity to purchase and utilize biodegradable films effectively [58].

**Table 3 polymers-16-03434-t003:** Summary of the biochar–bioplastic composite in the fabrication of mulch film.

Biochar Feedstock	Biochar Loading (wt %)	Base Polymer	Key Finding	Citation
Dairy manure Wood chip	10	PCL PLA	Biochar’s moisture content contributed to the hydrolytic degradation of the synthesized polymer.	[36]
Cassava rhizome Durian peel Pineapple peel Corncob	0.25	PLA	Carbon content in biochar improved mechanical properties (tensile elastic modulus and impact energy) of PLA/biochar composites.	[88]
Beechwood	5	PLA	Incorporating 5 wt% of biochar improved the composite’s tensile modulus of elasticity and strength.	[34]
Spent ground coffee	1, 2.5, 5, and 7.5	PLA	The content of BC highly influenced the ultimate properties of the PLA/BC biocomposites.	[89]
Switchgrass	12	PLA	Biochar significantly enhanced the hydrophobicity and mechanical characteristics relative to the control film.	[90]
Wood chips	10, 15, 20, and 30	PBAT/PLA	The degradation time of the composites was prolonged by a biochar content exceeding 15 wt%, which was attributed to the entrapment of PLA and/or PBAT within the matrix.	[38]
Post-consumer food waste	2.5, 5, 10, and 20	PBAT/PLA	The degradation rate of PLA was significantly increased by biochar under composting conditions, resulting in a nearly doubled mass loss in samples with a high biochar content after 40 days compared to neat PLA.	[86]
Wood Sewage sludge	10, 20	PLA	The use of biochars in biocomposites resulted in a reduction in the mechanical characteristics and impact strength as compared to PLA.	[91]
Pelleted miscanthus straw	1, 2.5, 5	PBS	The disintegration rate of biocomposites through enzymatic hydrolysis increased as the biochar content increased.	[35]
Birch and beech wood	5, 10, 20	PBAT	The elastic modulus was improved by biochar, while the deformation values were maintained at a high level.	[87]
Carob waste	10 and 20	PBAT	The dispersion grade and compatibility of biochar particles within the PBAT matrix were outstanding.	[92]
Waste coffee grounds	10, 20, and 30	PCL	The modulus of elasticity and tensile strength were not significantly impacted by the addition of biochar, despite the fact that the elongation at break decreased.	[63]
Wood	50	PBAT	A comprehensive techno-economic analysis and life cycle assessment indicated that biochar is currently not a viable choice in film production.	[92]

PLA: poly (lactic acid); PBAT: poly butylene-adipate-co-terephthalate; PCL: poly (ɛ-caprolactone); PBS: polybutylene succinate.

## 5. Effects of Biodegradable Mulch Film on Soil Carbon Dynamic

Although BDM may exhibit properties similar to traditional mulches when employed as a surface barrier, their ultimate outcomes are markedly different. Traditional films must be extracted from the soil surface, whereas BDMs are intended to be incorporated into the soil and decomposed by microorganisms. BDM has the ability to decompose in the soil system; thus, it may directly affect soil carbon dynamics [6].

Figure 3 shows the possible pathway of soil carbon stock contributing from the biochar incorporated with BDM after its deterioration. Biofilm formation is the initial step involving the development of a microbial community on the BDM surface via the release of extracellular polymeric molecules [36]. The enzymatic activity following biofilm development is the primary contributor to the next stage, which is depolymerization [93,94]. Depolymerization facilitates the disintegration of polymer chains into smaller molecules, including oligomers, trimers, dimers, and monomers, through the activity of extracellular enzymes. Subsequently, low-molecular-weight compounds, including dimers and monomers, are metabolized via transportation across the cell membrane, which is called bioassimilation [7]. Finally, mineralization, or complete biodegradation, denotes the breakdown of polymer fragments into mineralized constituents and biomass with the production of CO_2_ and H_2_O under aerobic circumstances [7,95]. The mulch fragments in the soil are subsequently converted by microbial activity into CO_2_ and microbial biomass. A fraction of the carbon from BDMs is assimilated into living microbial biomass, converting it into necromass upon the death of the microorganisms [68]. This material can additionally generate mineral-associated organic matter or be contained inside soil aggregates, thereby becoming persistent SOC [96]. Consequently, carbon obtained from BDMs can convert into stable SOC, potentially sequestering that carbon for an extended duration [68], as presented in Figure 3. Effective management of soil carbon on agricultural land is essential for sustainable crop production and the maintenance of soil ecosystem processes. Two crucial components of SOC, labile organic carbon and refractory organic carbon, are vital for the cycling and sequestration of organic carbon in soil [97]. According to reports, BDMs can have a direct impact on SOC pools by releasing carbon into the soil [68]. BDM regulations stipulate that 90% of the organic carbon in plastics, either in relation to the absolute amount of organic carbon or a control substance, must be converted to CO_2_ in standardized laboratory tests [68]. It is anticipated that as much as 10% of the carbon from BDMs may be converted into stable SOC annually. This indicates a possible enhancement in soil carbon stock of roughly 7.3 g C m^2^ after 5 years, 14.6 g C m^2^ after 10 years, and 29.2 g C m^2^ after 20 years of sustained application, presuming a standard mulch weight and carbon concentration [98]. Microbial necromass carbon constitutes 40%–55% of total soil carbon, predominantly derived from fungal necromass carbon, which accounts for 75% of this fraction. The microbial necromass carbon increased with the application of biodegradable materials, as the degradation of the film offered a more accessible substrate for microbes, resulting in enhanced microbial proliferation and, subsequently, an increase in microbial necromass carbon [8]. The addition of readily available C substrate with BDM might have caused the positive priming in soils. However, traditional polyethylene plastic films exhibited a negative correlation between the priming effect and mineralization in Vertisol soil during the incubation period at both 20 °C and 30 °C, as well as in Ferralsol soil at 20 °C [14,95].

Previous studies have proven the features of biochar that contribute to soil carbon. Biochar possesses the potential to serve as a crucial and readily available resource for sustainable agriculture, since it may effectively trap substantial amounts of carbon in soil over time, enhancing soil fertility, increasing crop output, and alleviating global warming [99]. Microorganisms can readily inhabit the biochar surface, providing an excellent environment and a supply of labile carbon and mineral nutrients [100]. The accumulation of bacterial and fungal necromass carbon is also affected by soil characteristics. Increased soil moisture and nutrient availability facilitate the accumulation of both fungal and bacterial necromass [8]. The incorporation of carbon-rich materials into the soil, including biochar, may facilitate the release of carbon, thus augmenting SOC pools [101,102]. The incorporation of biochar into rice paddy soil can enhance SOC by an average of 39%. This rise is markedly greater than that observed in conventional techniques, such as conservation tillage and cover cropping, which typically provide SOC increases of 6–8% [103]. Long-term effects of biochar revealed that its application might increase native soil organic carbon storage by 44–242% in macro-aggregates after three years, demonstrating significant enduring advantages [41]. Biochar generally consists of 62.2% to 92.4% carbon, varying with the feedstock types and synthesis parameters [104]. If we conservatively estimate that carbon in biochar is released into the soil following the degradation of BDMs, the carbon buildup in the soil could reach higher levels than BDMs alone. Thus, with the carbon contained in biochar-incorporated BDMs, the carbon from biochar would also be released into the soil as a carbon reservoir. However, the amount of the biochar dosage in the BDMs (Table 3) is extremely influenced by the carbon stock in the soil.

Biochar is becoming recognized as a viable approach for addressing climate change and improving soil carbon storage [51]. The degradation of biochar produced from ryegrass via compound-specific ^14^C was analyzed. The results indicate an extraordinarily sluggish disintegration rate, with the biochar depleting merely 7 × 10^−4^% of its carbon content daily under optimum conditions [105]. This indicates that it would require about 400 years for the biochar to undergo a simple 1% decrease in its carbon content [51]. These findings present compelling data that substantiate the persistent efficacy of biochar as a carbon sink, affirming its potential as a sustainable and durable solution for soil carbon sequestration. A fraction of the carbon from BDM that is assimilated into living microbial biomass would convert into necromass following the death of microorganisms. BDM material can additionally generate mineral-associated organic matter or be contained inside soil aggregates, thereby becoming persistent soil organic carbon. Consequently, carbon obtained from plastic can convert into stable soil organic carbon, potentially sequestering that carbon for an extended duration. Consequently, prolonged utilization of biodegradable plastic mulch may enhance soil carbon reserves, thereby improving soil health [68].

Biochar is recognized for its durability in soils, efficiently sequestering carbon for extended durations. The utilization of biochar in mulch films not only enhances soil carbon reservoirs but also decreases greenhouse gas emissions linked to conventional plastic mulches [106]. The integration of biochar with biodegradable mulch has demonstrated a substantial reduction in the carbon footprint of industrial systems. The incorporation of BDMs with carbon sequestration technologies has substantial environmental advantages, especially in improving soil health and alleviating climate change effects. This is a comprehensive summary derived from recent research. BDMs contain organic carbon, typically containing 60–80% carbon. When BDMs decompose, they contribute to soil organic matter, influencing biogeochemical cycling and potentially increasing soil carbon stocks. The studies by Zhou et al. [107] suggest that BMF can contribute approximately 0.30 tons of carbon per hectare per year to the soil, which complements other organic inputs like crop residues and root systems. Menossi et al. [66] also reported that BDMs contribute to soil organic matter, helping to sequester carbon and mitigate climate change impacts. After the fragmentation phase, microflora transforms the residual breakdown products of BDMs into carbon dioxide, methane, water, or biomass through the mineralization process, without harm. BDMs composed of biochar can efficiently break down in situ, reintegrating organic matter into the soil without producing detrimental leftovers. The decomposition process is enhanced by microbial activity, which transforms leftover components into innocuous byproducts such as carbon dioxide and water [66]. BDMs are incorporated into the soil at the conclusion of the growing season, adding physical fragments and a carbon source, as well as other constituents of plastic films (additives, plasticizers, minerals, etc.) that may further affect soil communities and their processes [6]. Research indicates that biodegradable mulches possess a high organic carbon content. Their incorporation into the soil enhances carbon storage and elevates organic carbon levels [108]. Soil microbes utilize the carbon from PBAT to derive energy, thus augmenting the soil’s carbon store [13]. Furthermore, the increase in warmth and humidity due to mulching may facilitate the mineralization of organic carbon in the soil, and studies suggest that mulching expedites the decomposition of soil organic carbon during the latter phases of crop development [12]. Although numerous reports state that mulching can increase the soil’s organic carbon contents, some claim that there is a decrease [8]. While biochar is considered a carbon-negative material, its use as a filler can lead to an increase in certain environmental impacts. For example, formulations using biochar were found to have a slightly higher global warming potential (3% increase) and a substantial impact on land use (+339%) compared with traditional fillers [54]. The disparity in carbon footprints between plastic films and BDM settings illustrates variations in the production processes of plastic films. The overall greenhouse gas emissions from the manufacturing of standard polyethylene amount to 2590 kg CO_2_-equivalent per hectare. The manufacture of polyethylene utilizes more fossil energy than biodegradable mulch, which decomposes entirely into water and carbon dioxide through the activity of environmental microorganisms and can be tilled directly without hand removal post harvest [58].

## 6. Challenges and Future Perspective

Biochar composites combined with biodegradable plastic mulches present a possible alternative to conventional plastic film mulches. Nonetheless, significant gaps exist in the comprehension of the long-term impact of biochar-BDMs on soil ecosystems essential for crop productivity. The impact of conventional mulches on soil microclimate, microbial populations, and biogeochemistry offers insight into the potential indirect effects of biodegradable mulches on soil [6]. The subsequent sections delineate the issues associated with the application of biochar integrated with BDM in agroecosystem scenarios.

### 6.1. Variability in Biochar Properties

The characteristics of biochar can differ markedly based on the feedstock employed and the manufacturing methodology, including pyrolysis temperature and resident time. This diversity can influence the efficacy of biochar in BDM applications, resulting in uneven material properties such as surface area, porosity, and chemical activity [109,110].

### 6.2. Processing and Functionalization

The necessity for efficient functionalization techniques to improve the compatibility of biochar with polymers presents a considerable challenge. Existing approaches may entail intricate chemical procedures that could raise further environmental issues, necessitating the development of more straightforward, environmentally sustainable functionalization strategies [111,112].

### 6.3. Mechanical Properties

Although biochar can enhance specific mechanical properties of BDMs, attaining optimal performance without sacrificing biodegradability presents a difficulty. Excessive filler content can result in brittleness in composites, thus restricting their practical applications [42,113].

### 6.4. Soil Ecosystem

Biochar can affect soil microbial communities in both beneficial and detrimental ways. Although it frequently improves microbial activity and diversity, some forms of biochar may release toxic compounds: certain biochars contain detrimental compounds, including polycyclic aromatic hydrocarbons (PAHs), which can negatively impact microbial communities and general soil health [114,115]. They may also alter nutrient cycling: the incorporation of biochar can alter the dynamics of nutrient availability and cycling in the soil, potentially resulting in adverse effects, such as heightened CO_2_ emissions from organic matter mineralization [115,116].

### 6.5. Incomplete Degradation

The challenges related to the total deterioration of BDM are substantial and complex. Although BDM composites with natural material filler (including biochar) provide a sustainable substitute for conventional plastic alternatives, their full decomposition is impeded by several environmental, material, and handling concerns. Some BDMs do not degrade as quickly or completely as expected under field conditions. Confronting these difficulties through research, innovation, and enhanced agricultural practices is crucial for optimizing their advantages while mitigating any adverse effects on soil health and the environment.

### 6.6. Economic Challenges

The manufacturing cost of biodegradable mulch films frequently exceeds that of conventional plastic films, presenting an economic obstacle for farmers contemplating their adoption. The cost disparity constitutes a substantial obstacle to adoption, particularly in areas where financial factors significantly impact agricultural choices [117]. To promote wider adoption, it is recommended that the cost of biodegradable films should not surpass 1.37 times that of traditional alternatives to guarantee a fair economic return for farmers [117,118].

Future research should focus on developing innovative functionalization methods that improve the interaction between biochar and biodegradable polymers while maintaining environmental safety. This could include bio-based modifications that enhance mechanical properties without introducing harmful chemicals. Implementing standardized production processes for biochar may facilitate uniform quality and efficacy in BDM applications. This entails tailoring pyrolysis settings unique to various feedstocks to generate biochars with favorable properties for composite applications. Further comprehensive research is required to comprehend the long-term environmental effects of biochar utilization in BDMs, especially for soil health and ecosystem dynamics. This involves evaluating the impact of various biochars on soil microbiomes and nutrient cycling when utilized as components of biodegradable materials.

Significant gaps persist in the comprehension of biochar-BDMs composite and their effects on soil ecosystems. Long-term research is required to evaluate soil health and sustainability implications, specifically concerning soil carbon and chronic toxicity effects, to fill these knowledge gaps. Furthermore, research should incorporate a direct comparison of traditional polymer and BDMs to ascertain whether BDMs influence soils differently than traditional plastic mulches. Numerous researchers suggest that BDMs may enhance decomposition; nevertheless, their impact on soil nutrient biogeochemistry remains largely unexamined [6]. Rectifying these knowledge deficiencies will furnish essential information to cultivators and regulators on the safety and sustainability of BDMs inside agroecosystems.

## 7. Conclusions

Biochar has a direct effect on the characteristics of synthesized BDM. The enhancement of mechanical properties and increasing the breakdown rate are mentioned in the literature. Frequent incorporation of biochar-BDM composite pieces into the soil may modify the soil’s physical environment and serve as a novel carbon source for microorganisms. While the overall carbon contribution from BDMs is minimal, their stimulatory effect on microbial activity may increase soil microbial biomass and, subsequently, soil organic matter. Nevertheless, substantial gaps remain in the current understanding of the effects of continuous BDM consumption on soil carbon stock. To address these knowledge gaps, long-term research is necessary to assess soil health and sustainability consequences, particularly regarding soil carbon impacts. Prolonged field trials are necessary to measure greenhouse gas emissions (the CO_2_ equivalent of methane, nitrous oxide, and alterations in soil organic carbon).

The future prospects for biochar integrated into BDM are favorable, propelled by innovations in agricultural methodologies and heightened environmental consciousness. The following are the principal factors influencing this perspective.

Notwithstanding the optimistic perspective, obstacles persist, such as elevated manufacturing expenses and fluctuating environmental circumstances that could influence performance. Moreover, additional research is required to comprehend the long-term effects of biochar utilization in mulch applications across various agroecosystems. Confronting these issues will be essential for optimizing the potential advantages of this integrated approach.

Promoting biodegradable films encounters numerous challenges, namely high costs, farmers’ reluctance to use them, and difficulties in their promotion. Continued research and technological innovations are anticipated to enhance the production methods of biochar and biodegradable mulch films. Innovations may result in the creation of more efficient formulations that optimize the advantages of biochar while preserving the beneficial characteristics of mulch films, including weed suppression and moisture retention.

Furthermore, extended field trials are necessary to measure greenhouse gas emissions (the CO_2_ equivalent of methane, nitrous oxide, and alterations in soil organic carbon). The carbon footprint associated with the inclusion of mulch microplastics and waste mulch should be regarded as a measurement indicator when examining the effects of various mulch treatments on carbon footprint.

## Figures and Tables

**Figure 1 polymers-16-03434-f001:**
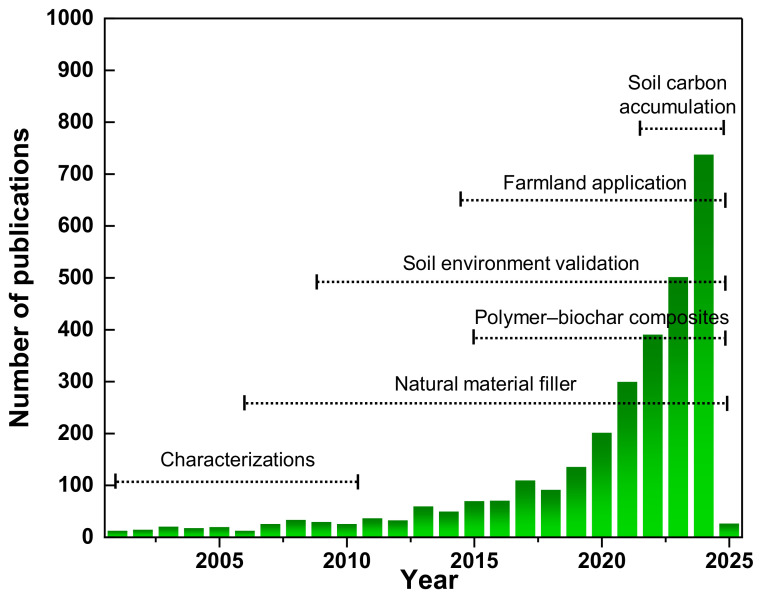
The quantity of publications from Science Direct pertaining to the keyword “biodegradable mulch film” from 2001 to 2025.

**Figure 2 polymers-16-03434-f002:**
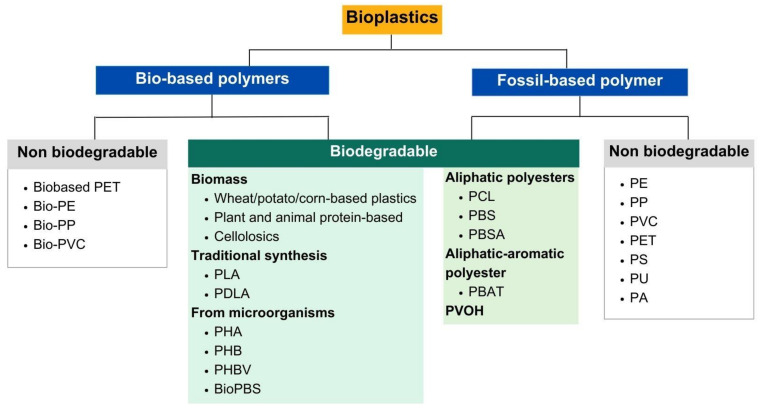
Biodegradation and source of bioplastic polymers. Adapted from [64]. PET: polyethylene terephthalate; PE: polyethylene; PP: polypropylene; PVC: polyvinyl chloride; PLA: poly(lactic acid); PDLA: (poly D-lactic acid); PHA: polyhydroxyalkanoate; PHB: poly (3-hydroxybutyrate); PHBV: poly (3-hydroxyvalerate); PVOH: poly(vinyl alcohol); PBAT: poly butylene-adipate-co-terephthalate; PBSA: poly(butylene succinate-co-adipate; PCL: poly (ɛ-caprolactone); PBS: polybutylene succinate; PS: polystyrene; PU: polyurethane; PA: polyamide.

**Figure 3 polymers-16-03434-f003:**
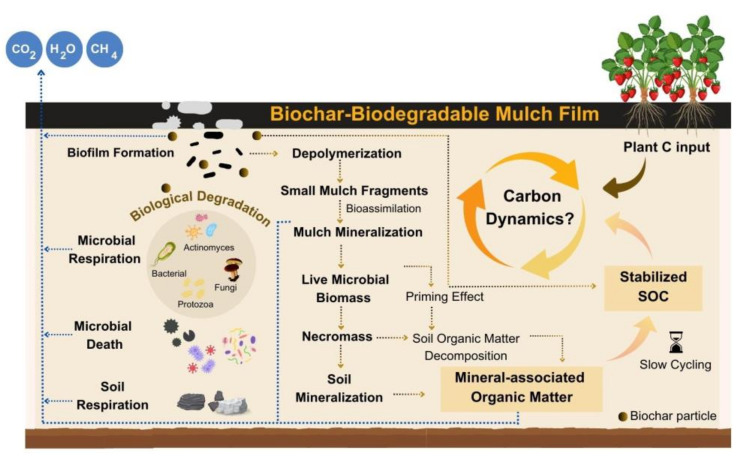
Effect of biochar biodegradable mulch film on soil carbon stock contribution, adapted from [68].

**Table 1 polymers-16-03434-t001:** Comparison of advantages and limitations of BDM with fossil-based non-biodegradable polymers.

Aspect	Bio-Based Polymers	BDM	Fossil-Based (Non-Biodegradable Polymers)
Source	Derived from renewable biological resources (e.g., plants)	Renewable biomass	Fossil fuels
Environmental Impact	Typically reduced carbon footprint	Biodegradation, mitigating plastic pollution	Non-biodegradable, contributes to long-term environmental pollution
Cost	Often more expensive due to production scale and technology maturity	Higher (due to raw materials and processing)	Lower (due to established fossil fuel supply chains)
Performance	May have varying mechanical properties; some are less durable than fossil-based options	Efficient in moisture retention and weed suppression	Highly effective for moisture retention and weed suppression
Market Adoption	Growing interest but still limited adoption in many sectors due to cost and performance concerns	Restricted market share (e.g., 5% in the USA) attributable to cost and awareness challenges	Prevalently utilized and endorsed in agriculture
Soil Impact	The degradation of polymers can serve as a carbon source for soil microorganisms	Augments soil microbial activity and vitality during decomposition	Adversely affect soil health and lead to microplastic pollution
Environmental Regulations	Complex and rapidly changing, driven by the need for sustainability in plastic use	Increasingly preferred in sustainable agricultural efforts	Confronting more stringent laws owing to environmental considerations
End-of-Life Options	Complex assessments needed to evaluate indirect land use change and other impacts	Composted or incorporated into soil without detrimental leftovers.	Typically ends up in landfills or burned, leading to pollution

**Table 2 polymers-16-03434-t002:** Summary of biodegradable polymer classification and their corresponding rates of biodegradation in soil.

Classification of Polymer	Polymer-Based Agricultural Mulches	Comparative Assessment of Biodegradation in Soil ^1^	Days to 50% of Biodegradation ^2^	Days to 90% of Biodegradation ^2^	Citation
Bio-based	Thermoplastic starch	High	30–60	60–120	[76,77]
	Chemically modified starch	High	30–90	90–180	[78]
	Cellulose	Moderately high	Not specified	63 days (late summer), 112 days (spring)	[79]
	PLA	Low	90	420	[80]
Fossil-based	PHB	Moderate	180	Not specified	[81]
	PHV	Moderate	180	Not specified	[81]
	PBAT	Low moderate	180	267	[82,83]
	PBSA	Low moderate	235	422	[82]
	PCL	Low moderate	Not specified	365	[84]
	PBS	Low moderate	495	890	[82]

PLA: poly(lactic acid); PHB: Poly (3-hydroxybutyrate); PHV: poly (3-hydroxyvalerate); PBAT: poly butylene-adipate-co-terephthalate; PBSA: poly(butylene succinate-co-adipate; PCL: poly (ɛ-caprolactone); PBS: polybutylene succinate. ^1^ Estimated comparative rate of biodegradation in soil is based on the literature [71]. ^2^ The degradation rate was assessed under soil conditions.

## Data Availability

Data available on request.

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
