# Peer review of "Does the Incorporation of Biochar into Biodegradable Mulch Films Provide Agricultural Soil Benefits?"

_polymers, 2024, doi:10.3390/polym16233434_

Round 1

Reviewer 1 Report

Comments and Suggestions for Authors

In my view the manuscript  may be considered for publication after the following modifications:

1-     Please compare BDM with other petroleum-based films toward identifying BDM’s advantages and disadvantages.

2-     In the introduction section, please present a clear explanation about BDM. 

3-     In the introduction section, please add examples about using biochar-derived biocomposite for producing BDM.

4-     In the introduction section, please explain the reason for using biochar composite-based BDMs for the soil carbon stock contribution.

5-     In the ‘’3’’ section, please present a comparison between two biodegradable polymer classification. You can also use articles in summary form in this section.

6-     Please add the ‘’Challenges and future perspective’’ section.

7-     In the conclusion section, please add future outlook.

Author Response

Thank you very much for your kind suggestions, please find the attached of my responses. 

Reviewer 2 Report

Comments and Suggestions for Authors

The manuscript is a review of biodegradable mulch film (BDM) materials and their features (primarily the effect of the filler on the physical and mechanical properties and biodegradation). Overall, this is a detailed review, but it leaves mixed feelings. The authors consider the problem and provide many references to various works, but it seems that this is only the beginning of some large work. Since this review is submitted to a thematic special issue (Advanced Biopolymers and Biocomposites), it may look good as an introduction to this special issue.

In any case, I would like to point out a number of comments to the authors:

1. Perhaps it is worth changing the title of the review - after all, most of it is devoted to BDM in general, and not to the effect of introducing biochar into them on carbon reserves in the soil.

2. In the context of modern research on biodegradable films, the problem of complete decomposition of such films is relevant, and not stimulation of decomposition to the level of microplastics. The authors mentioned the problem of microplastics a little, but they did not provide data for different fillers (including biochar) regarding their ability to cause complete decomposition of the film without turning into microplastics.

Author Response

Thank you very much for your kind suggestions, Please find the attached of my responses. 

Reviewer 3 Report

Comments and Suggestions for Authors

The review discusses the incorporation of biochar in biodegradable mulch (BDM) films and its effect on soil carbon stock. The structure of the review is suitable and the writing is good. The graphical representations in the review could help readers to know more details relevance to the study's findings. I think this review should be published in Polymers after minor revision:

(1) Please quote reference correctly for “Biochar possesses distinctive chemical, physical, and biological properties, making it a versatile material with a wide 101 range of applications.” in line 100-102 of page 2-3.

(2) Format: “2015” and “2020” in line 165 of page 4, “[14]” in line 170 of page 4, “[46]” in line 196 of page 5… should keep the same front of the manuscript. Please also check the whole manuscript.

(3) In the section 2-Quantitative assessment of the publications, could the authors give a brief description or explanation of the reason why “current investigations into the impact of biochar-enriched BDM on soil carbon have garnered less focus.” Is it because the BDM with biochar actually doesn’t have too much affect for the soil or environment and can be ignored? Please make the context more logical.

(4) In table 1, could authors give the comparable data (such as the needed time of 50% or full biodegradation) for different mulches, which would help the readers easily understand the difference.

Author Response

(The authors gave the same response as above.)

Round 2

Reviewer 1 Report

Comments and Suggestions for Authors

 Accept in present form